# Navigating the Rashomon Set: The Impact of Score Distributions and Decision Thresholds on Model Agreement

**Giovani Valdrighi**
Instituto de Computação
Universidade Estadual de Campinas
Campinas, Brazil

**Marcos Medeiros Raimundo**
Instituto de Computação
Universidade Estadual de Campinas
Campinas, Brazil

## Abstract

The existence of multiple equally accurate models for the same dataset, known as Rashomon effect, has recently attracted the attention of the machine learning community. The multiplicity of models permits practitioners to select for accurate models, but also satisfy other objectives such as fairness or interpretability. However, the disagreement of models on individual samples, measured by ambiguity, can reduce the credibility of decisions. In this paper, we theoretically study the ambiguity of models obtained from a randomized training procedure by relating it to the distribution of residuals, and more particularly, the occurrence of large residuals. Based on our theoretical results, we present a simple yet effective approach for threshold selection that reduces ambiguity at a low cost to accuracy. We also present an adapted loss for binary classification that reduces ambiguity by controlling the tail of the distribution of residuals. In experiments using five datasets, our methodology demonstrated a reduction in ambiguity at a low cost in terms of both accuracy and computational resources.

## 1 Introduction

Recent studies from *Predictive Multiplicity* have shown that common complex machine learning models, already widespread in many social applications, can present the *Rashomon effect*: a set of competing models that achieve the same accuracy on a dataset (Breiman, 2001; Marx et al., 2019; Hsu & du Pin Calmon, 2022). This effect enhances the effectiveness of unfairness mitigation techniques (Black et al., 2022) and addresses interpretability concerns (Rudin et al., 2024), as a large and possibly diverse set of accurate models allows practitioners to select models based on criteria other than performance. Previous work has studied properties of the set of good models, called the *Rashomon set*, such as how to properly sample models from it (Jain et al., 2025) and how its size relates to characteristics of the hypothesis space or the data generation process (Hsu & du Pin Calmon, 2022; Semenova et al., 2023). Yet despite the opportunities offered by multiple good models, disagreement among them can raise questions about their validity.

In practical deployments, a model $h$ is obtained from a procedure $\mathcal{T}$ applied to a dataset $\mathcal{D}$. Commonly, $\mathcal{T}$ will include randomized steps, such as splitting training and validation or initializing an algorithm. A secondary model $h'$ with similar accuracy could be obtained by simply repeating $\mathcal{T}$ with another random seed. Despite being equally accurate, models $h$ and $h'$ can disagree on a subset of samples from the dataset, as measured by the *ambiguity* rate (Marx et al., 2019). When the choice between $h$ and $h'$ is based on an unconscious decision to select a random seed, individuals in the disagreement set may end up with arbitrary decisions (Ganesh et al., 2025), which can erode trust in algorithmic decision-making. Previous work has presented algorithms for resolving conflicts among a large set of models obtained from retraining, such as reconciling (Roth et al., 2022; Behzad et al., 2025; Du et al., 2025) or using ensembles (Long et al., 2023; Hsu et al., 2024), to reduce ambiguity.

Despite these advancements, existing approaches, such as model reconciliation or ensembling, manage multiplicity by post-processing a set of models. However, these methods do not handle ambiguity in single models. In this work, we study how ambiguity can be bounded by properties of

models obtained by the training procedure. In this work, we study how ambiguity (even in single models) can be bounded by properties of models obtained by the training procedure. By focusing on the fact that all models are approximations to a shared function, we can bound ambiguity by how probable models can deviate from this reference, that is, present a large residual. Furthermore, we focus on the effect of the threshold on ambiguity, an important consideration in practical classifier implementations. Our theoretical results permit us to design and develop training procedures that provably constrain the ambiguity of single models by controlling the residual distribution, providing guarantees that hold prior to any model selection step.

## 2 NOTATION AND DEFINITIONS

We consider a classification task with a dataset $\mathcal{D} = \{(x_i, y_i)\}_{i=1}^n$ drawn from $P_{X,Y}$ where $x_i \in \mathbb{R}^d$ and $y_i \in \{0, 1\}$. The practitioner selects classifiers from a hypothesis space $\mathcal{H}$ where each model $h \in \mathcal{H}$ is a mapping $h : \mathbb{R}^d \to [0, 1]$; that is, $h$ outputs scores that try to approximate the reference $r(x) := P_{Y|X}(Y = 1 \mid X = x)$. Final decisions are made with a threshold $t \in [0, 1]$ by $f_t(h(x)) = 1\{h(x) > t\}$ where $1\{\cdot\}$ is the indicator function. We also write $[k] = \{1, \ldots, k\}$.

Given a loss measure $L(h, \mathcal{D})$ for model $h$ on data $\mathcal{D}$, such as the mean absolute error, the **Rashomon set** is defined as the set of models with almost-equal performance:

$$\mathcal{R}(\mathcal{H}, \mathcal{D}) := \{h \in \mathcal{H} \mid L(h, \mathcal{D}) \leq \epsilon\} \tag{1}$$

where $\epsilon \in \mathbb{R}_{>0}$ is a parameter that controls the size of the set. Commonly, $\epsilon$ is defined as the loss of the optimal model from $\mathcal{H}$ plus a small tolerance, which ensures that the set is not empty. Previous work has analyzed the size and diversity of the Rashomon set for different spaces $\mathcal{H}$; however, when considering highly complex models, it becomes infeasible to assess the complete Rashomon set, motivating the use of empirical approximations.

**Empirical Rashomon Set of a Randomized Training Procedure** Similarly to Long et al. (2023), we consider that models $h$ are obtained by a randomized procedure applied to the dataset, noted as $\mathcal{T}$. This procedure can include, for example, randomly splitting the dataset $\mathcal{D}$ into training and validation sets before training, or using a randomized hyperparameter search. Then, the empirical approximation of the Rashomon set with $m$ models is:

$$h_1, \ldots, h_m \overset{i.i.d.}{\sim} \mathcal{T}(\mathcal{D}) \quad \hat{\mathcal{R}}(\mathcal{T}, \mathcal{D}, \epsilon) := \{h_j \mid L(h_j, \mathcal{D}) \leq \epsilon \ \forall j \in [m]\}, \tag{2}$$

where $h \sim \mathcal{T}(\mathcal{D})$ express that $h$ is obtained from the random procedure applied to the dataset $\mathcal{D}$. When $m$ gets bigger, $\hat{\mathcal{R}}_m$ approximates the complete set of models $h \in \mathcal{H}$ that can be obtained by $\mathcal{T}(\mathcal{D})$ with loss below $\epsilon$.

We are interested in the common case in which models in $\hat{\mathcal{R}}_m$ produce conflicting predictions. To measure this, we consider the ambiguity of decisions.

**Definition 2.1** (Worst-case Ambiguity, adapted from (Marx et al., 2019)). Given a finite set of models $\mathcal{R} \subset \mathcal{H}$, and a threshold $t \in [0, 1]$, the worst-case ambiguity for a sample $x \in \mathbb{R}^d$ is:

$$A_{worst}(x) = \max_{h, h' \in \mathcal{R}} 1\{f_t(h(x)) \neq f_t(h'(x))\} \tag{3}$$

However, by measuring ambiguity with this approach, we obtain a pessimistic measure, since $A(x)$ can be 1 even with a single disagreeing model $h$ from a large set of models $\mathcal{R}$. Considering our empirical approximation of the Rashomon set, we will focus our study on the ambiguity between two models randomly selected from $\hat{\mathcal{R}}_m$.

**Definition 2.2** (Ambiguity). Given a finite set of models $\mathcal{R} \subset \mathcal{H}$ and a threshold $t \in [0, 1]$, and let $h \sim U[\mathcal{R}]$ mean that $h$ is randomly selected from $\mathcal{R}$, the pairwise ambiguity for a sample $x \in \mathbb{R}^d$ is:

$$A(x) = \mathbb{E}_{h, h' \sim U[\mathcal{R}]} [1\{f_t(h(x)) \neq f_t(h'(x))\}] \tag{4}$$

Another common metric of model multiplicity was introduced by Black et al. (2022). Called pairwise disagreement, instead of evaluating the disagreement of a fixed sample over randomly selected models, it evaluates the disagreement of fixed models over randomly selected samples. We formalize it in the following definition:

**Definition 2.3** (Pairwise disagreement, adapted from (Black et al., 2022)). Given a distribution of samples $P_X$, two fixed models $h, h'$ and a threshold $t \in [0, 1]$, the pairwise disagreement is:

$$d(h, h') = \mathop{\mathbb{E}}_{X \sim P_X} [1\{f_t(h(x)) \neq f_t(h'(x))\}] \tag{5}$$

In this work, we focus our attention on the average ambiguity over a dataset, that is, $\mathbb{E}_{X \sim P_X}[A(X)]$. However, we can relate the two introduced multiplicity metrics as follows:

**Proposition 2.4** (Connection between ambiguity and pairwise disagreement). *If $\mathcal{R}$ is a finite set of models, we can relate the ambiguity and pairwise disagreement by calculating expectation over $X$ and $h, h'$, respectively, as follows:*

$$\mathop{\mathbb{E}}_{X \sim P_X} [A(X)] = \mathop{\mathbb{E}}_{h, h' \sim U[\mathcal{R}]} [d(h, h')] \tag{6}$$

While the Rashomon set is based on the scoring models $h$, we are interested in studying the ambiguity of thresholded decisions, since the final binary decision determines the outcome for individuals. In this work, we investigate how ambiguity over a dataset $\mathcal{D}$ can be bounded from characteristics of models $h \in \hat{\mathcal{R}}_m$. On following sections, we do this by considering that models generated from the procedure $\mathcal{T}(\mathcal{D})$ are approximations to $P_{Y|X}(Y = 1|X = x)$. The quality of this approximation, thus, can be leveraged to understand ambiguity.

## 3 BOUNDING AMBIGUITY

This section presents our theoretical analysis of ambiguity, with all proofs presented in Appendix A. Our study is based on the generic consideration that models $h$ from the Rashomon set are approximations of the true distribution $r(x) := P_{Y|X}(Y = 1|X = x)$. This can be quantified by the random residual $|h(X) - r(X)|$ for a fixed $h \in \hat{\mathcal{R}}_m$. Furthermore, we consider the importance of the threshold $t$ in defining decisions. In this way, two models $h, h'$ will disagree if they are: 1) close to the threshold $t$ and present a small variation of scores, or 2) distant from the threshold and present a large disagreement on the scores. We formalize these ideas into a simple proposition:

**Proposition 3.1** (Average Ambiguity). *With a Rashomon set $\hat{\mathcal{R}}_m$, a threshold $t$ and a constant $\gamma \in \mathbb{R}_{>0}$, define the concentration measure $S_\gamma(t) := P_X(|h(X) - t| \leq \gamma)$ and let $C_\gamma := \max_{h \in \hat{\mathcal{R}}_m} P_X(|h(X) - r(X)| \geq \gamma)$. Then, the average ambiguity is bounded by:*

$$\mathop{\mathbb{E}}_{X \sim P_X} [A(X)] \leq 2S_\gamma(t) + 2C_\gamma \tag{7}$$

This result indicates that the average ambiguity is related to the residual of score models, as expressed by the constant $C_\gamma$. While in practice models are generally selected based on the performance of thresholded decisions, this result highlights the importance of having low-scoring decisions. The term $S_\gamma$ reflects the confidence of the model around a particular threshold, and models with a larger margin will present a lower bound on ambiguity. Furthermore, $S_\gamma$ also reflects a characteristic of the data distribution, as when models from $\hat{\mathcal{R}}_m$ are a perfect approximation, we can write $S_\gamma(t) = P_X(|r(X) - t| \leq \gamma)$, which only depends on $r(x)$ and the selected threshold $t$.

We present an illustrative example to showcase the relation between ambiguity and residuals in practice. We generated a synthetic dataset with a centered feature $X \sim Beta(p, p) - 0.5$, for a parameter $p \in \mathbb{R}_{>0}$ and $Y$ is obtained from a logistic model $r(x) = 1/(1 + \exp\{-6x\})$. To obtain models $h$, we create a simple training procedure that

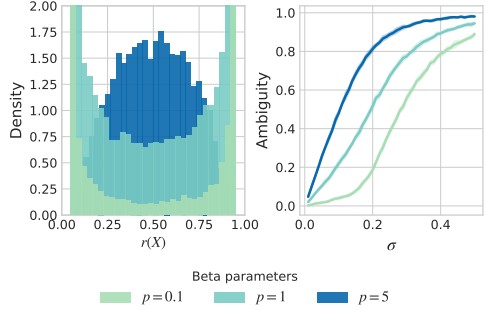

Figure 1: Ambiguity of a synthetic dataset where score models are generated by adding noise to the true scores with probability. Ambiguity increases when models deviate from the reference (larger $\sigma$) and when probabilities concentrate around $0.5$ (larger $p$).

outputs models $h(x) = r(x) + \mathcal{N}(0, \sigma)$, where $\sigma$ controls how close models are to the reference. Furthermore, when $p$ is smaller, the concentration of $S_\gamma(0.5)$ for the threshold $0.5$ is reduced. Fig. 1 displays the results for multiple configurations of parameters $p, \sigma$. By increasing $\sigma$, scores deviate from the reference, and the ambiguity increases from $0$ to almost $1$. Furthermore, data distributions with higher concentration around the threshold (larger $p$) will result in models with greater concentration and will increase ambiguity more quickly.

Our next result considers the probability of obtaining at least one conflicting prediction for a dataset of size $n$, extending the result from Prop. 3.1.

**Theorem 3.2** (Ambiguity of a Dataset). *With the same definitions as from Prop. 3.1, and let $\mathcal{D}_{test}$ be a dataset of $n$ samples that is independent of $\hat{\mathcal{R}}_m$, and let $h, h' \sim U[\hat{\mathcal{R}}_m]$. The probability of models $h, h'$ disagreeing on any of the samples from $\mathcal{D}_{test}$ is bounded, with probability $1 - \delta$, by:*

$$\Pr_{h, h' \sim U[\hat{\mathcal{R}}_m]} \left( \bigcup_{x \in \mathcal{D}_{test}} 1\{f_t(h(x)) \neq f_t(h'(x))\} \right) \leq n \left( 2S_\gamma(t) + 2C_\gamma + \sqrt{\frac{\ln(1/\delta)}{2n}} \right) \quad (8)$$

Following, we employ traditional results from learning theory to obtain more informative bounds on the residual $P_X(|h(X) - r(X)| \geq \gamma)$ and replace the generic constant $C_\gamma$. By applying the Markov inequality, we can relate ambiguity to the models' Brier scores. Notice that despite us using $|h(X) - r(X)|$, this quantity is unknown as we generally do not have access to $P_{Y|X}$, while the Brier score can be computed from collected data.

**Theorem 3.3** (Brier Score Bound for $C_\gamma$). *Let $BS(h) = \mathbb{E}_{X,Y}[(h(X) - Y)^2]$ be the Brier score of a model $h$. Under the definitions of Prop. 3.1, the constant $C_\gamma$ is bounded by the maximum Brier score in the $\hat{\mathcal{R}}_m$, scaled by $\gamma^2$:*

$$C_\gamma \leq \frac{1}{\gamma^2} \max_{h \in \hat{\mathcal{R}}_m} BS(h) \quad (9)$$

From this result and Prop. 3.2, we can relate the Brier score, a known measure of calibration and event uncertainty, to the ambiguity of models from the Rashomon set. While the Brier score offers a convenient proxy, it fundamentally constrains the average mass of residuals. However, $C_\gamma$ consists of a tail bound. A quadratic penalty might tolerate rare but severe deviations that we want to avoid. To constrain ambiguity strongly, it is important to investigate bounds/strategies that control the probability of large errors more aggressively than the second moment allows.

**Theorem 3.4** (Chernoff Bound for $C_\gamma$). *Under the definitions of Prop. 3.1, for any parameter $s > 0$, the constant $C_\gamma$ is bounded by the expected exponential absolute error against the labels:*

$$C_\gamma \leq e^{-s\gamma} \mathbb{E}_{X,Y} \left[ e^{s|h(X) - Y|} \right] \quad (10)$$

This result establishes a link between the ambiguity of a dataset and the *exponential loss* of the models in the $\hat{\mathcal{R}}_m$. Unlike the Brier score (Theorem 3.3), which penalizes deviations quadratically (MSE), the Chernoff bound suggests that ambiguity is bounded by a loss that penalizes deviations exponentially (and intrinsically penalizes higher moments). In practice, to minimize arbitrariness, a training procedure should not only minimize average error but also emphasize samples with the highest residuals.

**Theorem 3.5** (Bennett's Bound for $C_\gamma$). *Under the definitions of Prop. 3.1, let $M$ be an upper bound for residuals of any $h \in \hat{\mathcal{R}}_m$ ($|h(X) - Y| \leq M$ always), and let $\mu = \mathbb{E}_{X,Y}[|h(X) - Y|]$ be the mean absolute error. For any $\gamma > \mu$, the constant $C_\gamma$ is bounded by:*

$$C_\gamma \leq \exp \left( -\frac{BS(h)}{M^2} g \left( \frac{M(\gamma - \mu)}{BS(h)} \right) \right) \quad (11)$$

*where $g(u) = (1 + u)\ln(1 + u) - u$.*

This result refines the upper bound on ambiguity by decomposing the error into its mean $\mu$, its Brier score $BS(h)$, and its scale $M$. As discussed in Remark A.2, this decomposition reveals that minimizing the Brier score is the most effective practical lever, inducing a fast polynomial decay in ambiguity. However, the presence of $M$ in the exponent indicates that higher-order moments, specifically the worst-case residual, remain significant. Thus, while our primary optimization focus is on the second moment, the bound suggests that a robust strategy must also implicitly constrain the scale of extreme deviations represented by $M$.

## 4    DEALING WITH MULTIPLICITY

In this section, we leverage our theoretical results to adapt the training procedure $\mathcal{T}$ to obtain Rashomon sets with lower ambiguity. While previous works have presented how to reduce ambiguity by updating/aggregating a set of $m$ trained models (for example, Long et al. (2023) creates an ensemble, Roth et al. (2022) performs reconciliation), our methodology alters the optimization landscape of $\mathcal{T}$ to penalize regions of high potential ambiguity, integrating the solution directly into the learning process rather than treating it as a post-processing step.[1]

### 4.1    THRESHOLD SELECTION

While previous work has considered decision ambiguity and not predicted scores, it analyzed it using a fixed threshold of 0.5. However, Prop. 3.1 shows that the selected threshold and its interaction with the distribution of scores present an influence on ambiguity.

In a standard development pipeline, after obtaining a scoring model $h \sim \mathcal{T}(\mathcal{D})$, a threshold $t$ is selected by maximizing a performance metric, such as accuracy. We present a simple modification to the threshold selection phase that reduces ambiguity at a small cost to accuracy. As shown by our proposition, a smaller concentration $S_\gamma(t)$ will reduce the upper bound of ambiguity. Let $ACC(t) := \sum_{(x,y)\in\mathcal{D}} 1\{f_t(x) = y\}$ be the accuracy obtained from a threshold $t$, and let's define $t^\star := \arg\max_t ACC(t)$. Given a tolerance $\delta$, we select the threshold by solving the optimization problem:

$$\min_t \ S_\gamma(t) \quad \text{s.t. } ACC(t) \geq ACC(t^*) - \delta \tag{12}$$

This procedure will select the threshold that presents the minimal concentration $S(t)$, while only decreasing accuracy by $\delta$. As our experiments at Sec. 5 show, this simple procedure presents significant gains.

### 4.2    MINIMIZING AMBIGUITY VIA TILTED OBJECTIVE

Our secondary proposed modification to the training procedure $\mathcal{T}$ is directly motivated by the theoretical results in Theorem 3.4. The theorem establishes that the ambiguity of a Rashomon set is upper-bounded by the exponential moment of the residuals $|h(x) - y|$. Consequently, to control for ambiguity, we must minimize this exponential expectation rather than the standard average error. Let the residual be $\ell(x, y) := |h(x) - y|$, we define the training objective as:

$$\mathcal{L}(\mathcal{D}) := \frac{1}{s} \ln \left( \frac{1}{n} \sum_{(x,y)\in\mathcal{D}} e^{s\ell(x,y)^2} \right) \tag{13}$$

where $s > 0$ is a hyperparameter. As $s \to 0$, this objective recovers the expectation of $\ell(x, y)^2$ that consists of the Brier score, whereas $s \to \infty$ approaches a min-max objective, focusing exclusively on the worst-case sample.

This formulation is similar to Tilted Empirical Risk Minimization (TERM) (Li et al., 2021) and Entropic Value at Risk (EVaR) (Ahmadi-Javid, 2012). However, while previous work has primarily motivated TERM through the lenses of robust generalization or fairness, our work is the first to relate it to predictive multiplicity. Also, as posed in Theo. D.1, we show that Entropic Value at Risk (EVaR) is valid in any moment $E[\ell(x, y)^k]$ of $\ell(x, y)$, which helped us to find a more numerically stable approach when $k = 2$.

Worth mentioning that while Theorem 3.5 bounds ambiguity via the exponential of absolute errors ($k = 1$), we leverage the Generalized EVaR (Theorem D.1), which shows equivalence to any moment, to optimize the second moment ($k = 2$). This retains the theoretical guarantees while providing superior numerical stability regarding the variance of residuals.

At Sec. 5, we empirically show the optimization of this objective using stochastic gradient descent.

---

[1]At our experiments, repeated execution of $\mathcal{T}(\mathcal{D})$ was necessary to measure ambiguity; however, this is not necessary in practical deployments.

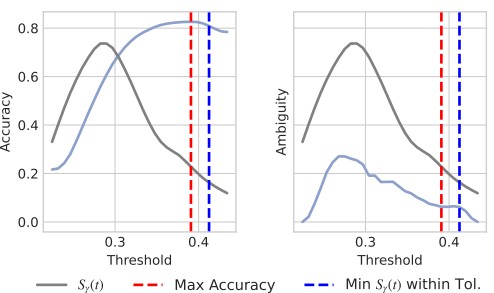 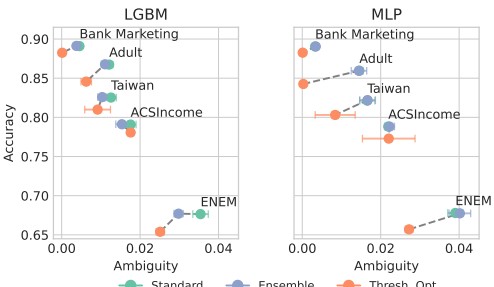

Figure 2: Accuracy, ambiguity, and concentration of LGBM with the Taiwan dataset for multiple thresholds. Red line indicates the threshold that maximizes accuracy, and the blue line indicates the threshold obtained from Eq. 12.

Figure 3: Accuracy and ambiguity of two training procedures that only differ in the methodology for threshold selection. *Thresh. Opt.* procedure select thresholds to minimize concentration score $S_\gamma(t)$, reducing ambiguity.

## 5 EXPERIMENTS

We present a set of experiments to showcase the practical implications of the proposed methodologies from Sec. 4 in terms of accuracy and ambiguity. Experiments have the objective of answering whether reduced ambiguity can be obtained from: 1) a simple adaptation of the methodology of threshold selection, and 2) by the use of the tilted loss.

### 5.1 EXPERIMENTS DETAILS

**Models**  We consider MLP and gradient-boosting models (LightGBM implementation). MLPs are implemented using PyTorch and optimized with stochastic gradient descent. Despite this selection of models, resulted from Sec. 4, are generic and could be incorporated with other models and learning algorithms.

**Datasets**  We use five common benchmark datasets for classification in the finance, social, and marketing domains. Datasets were split into 80% training and 20% test, with the training data further split 80%\20% for validation during the randomized procedures. All datasets followed a similar preprocessing methodology: replacing missing values with a flag, one-hot encoding for categorical variables, and scaling numerical variables. Appendix B presents a more detailed description of datasets.

**Procedures**  We consider two main types of randomized procedure: 1) randomized split of training (80%) and validation (20%) with default hyperparameters of the algorithm, written as $\mathcal{T}_{split}$, and 2) a similar procedure, but with 10 trials of hyperparameter tuning using Optuna Akiba et al. (2019), write as $\mathcal{T}_{tune}$. Training procedures do not include a threshold selection method; the default approach is to select the threshold that maximizes accuracy. To obtain $\hat{\mathcal{R}}_m$, we execute the randomized procedures 10 times with different random seeds, which are then used to calculate ambiguity and the average accuracy of models in the set. We repeat all experiments 10 times with different random seeds to compute the standard deviation of the results.

### 5.2 THRESHOLD SELECTION

We compare the threshold selection methodology presented at Sec. 4.1 with the standard approach of selecting the threshold that maximizes accuracy on validation. The study is done with the $\mathcal{T}_{split}$ training procedure.

**Analysis at Multiple Thresholds**  At Fig. 2, we present accuracy and ambiguity obtained at multiple thresholds from a set of LGBM models fitted in the Taiwan dataset. As the threshold increases, accuracy rises to 82%. Interestingly, ambiguity decreases from the maximum of 0.4 to around 0.1 in the threshold that maximizes accuracy. It can be seen that the region of higher ambiguity is also

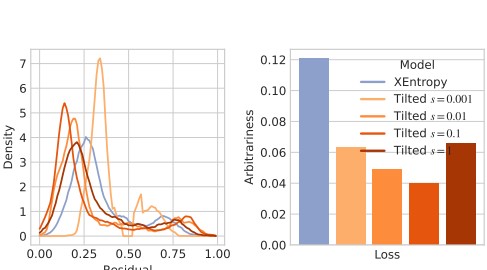 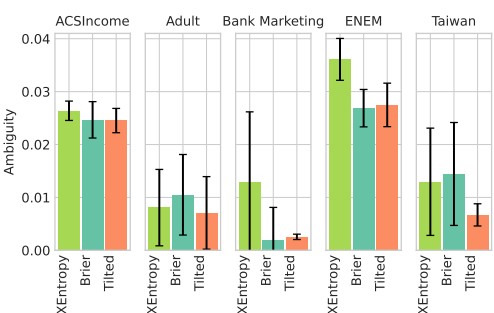

Figure 4: Effect of tilted loss in ambiguity with an MLP in the Taiwan dataset.

Figure 5: Average ambiguity of different loss functions with an MLP using the procedure $\mathcal{T}_{tune}$.

the region with the highest concentration $S_\gamma(t)$, as indicated by Prop. 3.1. Lower ambiguity can be achieved by selecting a threshold near the one that maximizes accuracy. By allowing a 2.5% decrease in accuracy, the blue threshold can be selected, reducing ambiguity.

**Comparative Analysis**  Fig. 3 displays the results of the threshold selection methodology proposed for all datasets and two types of models (MLP and LGBM) using the $\mathcal{T}_{split}$ procedure. Except for the ACSIncome dataset, the threshold selection methodology reduced ambiguity compared with the threshold that maximizes accuracy, yielding almost 0% ambiguity for the Bank Marketing dataset with models. Results for an ensemble (10 models) are also presented for comparison, but the reduction in ambiguity is smaller than that achieved by the threshold selection approach. The procedure was executed with an acceptable loss of accuracy of at most 2.5%; however, all scenarios resulted in a lower cost of accuracy than the tolerance.

## 5.3  EVALUATING TILTED LOSS

**Hyperparameter and Residual Distributions**  We evaluate the hyperparameter $s \in \{0.001, 0.01, 0.1, 1\}$ from the tilted loss (Eq. 13) and compare it to standard binary cross-entropy using MLP in the Taiwan dataset using the procedure $\mathcal{T}_{split}$. Fig. 4 presents the distribution of residuals and the ambiguity obtained. By shifting the objective from binary cross-entropy to the tilted objective, ambiguity decreases significantly. We see that by increasing the hyperparameter value from 0.001 up to 0.1, there is a decrease in ambiguity. Similarly, the tail of residuals can be seen to be reduced. However, when using $s = 1$, ambiguity increases again. This might occur due to the difficulty in ensuring numerical stability of the tilted loss, which might reach large values when the hyperparameter increases.

**Multiple Losses with Hyperparameter Tuning**  To evaluate the tilted loss in a more realistic scenario, we perform a larger experiment using the $\mathcal{T}_{tune}$ procedure (which includes hyper-parameter optimization) on all datasets. We compare the tilted loss (Eq. 13) with $s = 0.1$ against using standard cross-entropy and using the Brier score as a loss measure (inspired by Theo. 3.3). Results are displayed by selecting the threshold that maximizes accuracy on the validation data. Results are displayed at Tab. 1 and ambiguity is visualized in Fig. 5. The tilted loss showed the lowest ambiguity across all datasets, except for BankMarketing. Interestingly, using the Brier score instead of cross-entropy also reduced ambiguity in the ACSIncome, BankMarketing, and ENEM datasets, as shown in Theo. 3.3. As shown in Tab. 1, the Tilted loss also does not present a significant decrease in accuracy.

**Sensitivity Analysis of Rashomon Set**  To assess the effectiveness of the tilted loss, we evaluate its sensitivity to different sizes of the Rashomon set. We evaluate different definitions of $\hat{\mathcal{R}}_m$ by selecting models that have loss $L \leq (1 + \epsilon)L_{\min}$ for $\epsilon \in \{0.01, 0.02, 0.03, 0.04, 0.05\}$ where $L_{\min}$ be the minimal loss of a set of models. By increasing $\epsilon$, we include models with larger performance differences and, consequently, greater diversity. We leverage the $\mathcal{T}_{tune}$ procedure with an MLP using binary cross-entropy and the tilted loss. Thresholds are selected to maximize accuracy. Results are

Table 1: Comparison of model performance and ambiguity.

| Dataset | Loss | Acc ($\uparrow$) | Ambiguity ($\downarrow$) |
| --- | --- | --- | --- |
| ACSIncome | XEntropy | 0.772 ($\pm$ 0.0) | 0.026 ($\pm$ 0.0) |
| ACSIncome | Brier | 0.772 ($\pm$ 0.0) | 0.025 ($\pm$ 0.0) |
| ACSIncome | Tilted | **0.776 ($\pm$ 0.0)** | **0.025 ($\pm$ 0.0)** |
| Adult | XEntropy | 0.762 ($\pm$ 0.01) | 0.008 ($\pm$ 0.01) |
| Adult | Brier | 0.762 ($\pm$ 0.0) | 0.011 ($\pm$ 0.01) |
| Adult | Tilted | 0.762 ($\pm$ 0.01) | **0.007 ($\pm$ 0.01)** |
| BankMarketing | XEntropy | 0.739 ($\pm$ 0.01) | 0.013 ($\pm$ 0.01) |
| BankMarketing | Brier | 0.747 ($\pm$ 0.01) | **0.002 ($\pm$ 0.01)** |
| BankMarketing | Tilted | **0.881 ($\pm$ 0.01)** | 0.003 ($\pm$ 0.0) |
| ENEM | XEntropy | 0.678 ($\pm$ 0.0) | 0.036 ($\pm$ 0.0) |
| ENEM | Brier | 0.68 ($\pm$ 0.0) | **0.027 ($\pm$ 0.0)** |
| ENEM | Tilted | **0.679 ($\pm$ 0.0)** | **0.027 ($\pm$ 0.0)** |
| Taiwan | XEntropy | 0.758 ($\pm$ 0.01) | 0.013 ($\pm$ 0.01) |
| Taiwan | Brier | 0.76 ($\pm$ 0.0) | 0.014 ($\pm$ 0.01) |
| Taiwan | Tilted | **0.807 ($\pm$ 0.0)** | **0.007 ($\pm$ 0.0)** |

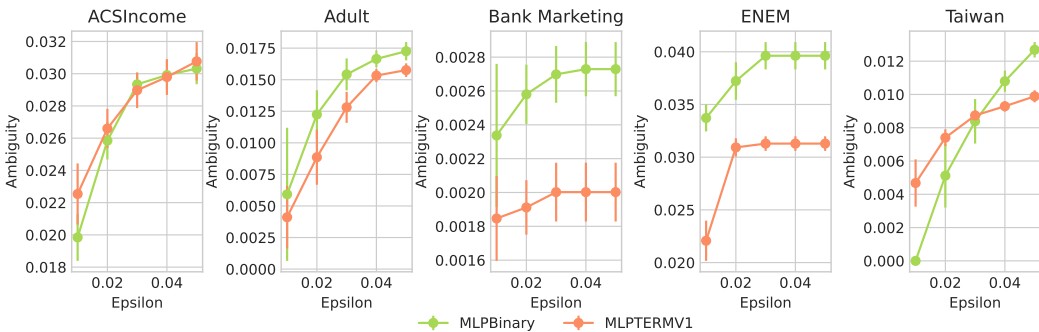

Figure 6: Ambiguity calculated over different configurations of the Rashomon set.

displayed in Fig. 6. By increasing the size of the Rashomon set, ambiguity increases in all of the datasets presented. According to Prop. 3.1, introducing extra models to $\hat{\mathcal{R}}_m$ will only increase the constant $C_\gamma$. However, when we compare the tilted loss to the standard binary loss, ambiguity is more controlled with our approach, presenting a smaller increase than the standard loss.

## 6 RELATED WORKS

In this section, we situate our contributions within the literature on predictive multiplicity and variance-regularized optimization. For a comprehensive systematization of predictive multiplicity, see Ganesh et al. (2025).

**Predictive Multiplicity**   Previous work has focused on how to measure the occurrence of predictive multiplicity and on the properties of the Rashomon set. For example, the *Rashomon capacity* proposed by Hsu & du Pin Calmon (2022) (variation of predicted scores among samples) or the *Rashomon ratio* proposed by Semenova et al. (2023) (fraction of models from the hypothesis space that are inside the Rashomon set). Following these measures enabled the study of the Rashomon set and its relationships with the hypothesis space, data distribution, and other aspects.

A different set of works has focused on reducing predictive multiplicity. More related to our work, Long et al. (2023) presented a bound for ambiguity generated by ensemble models, which permitted to bound $A(x)$ independently of $x$. In contrast, our work derives a bound over the distribution of $x$. Similarly, Hsu et al. (2024) leverages ensembles to control for multiplicity in gradient-boosting models. An alternative to ensembling is model reconciliation. Roth et al. (2022) proposed *Reconcile*,

an algorithm that merges two models into a new one by updating predictions of conflicting samples. This procedure was empirically evaluated by Behzad et al. (2025), studying the reconciliation of a large set of models.

**Robust Optimization**   Our work also relates to optimization frameworks that modify standard loss measures to achieve robustness. Li et al. (2021) introduced Tilted Empirical Risk Minimization (TERM), a general framework that applies an exponential tilt to standard losses, which can interpolate between standard empirical risk minimization and min-max optimization. Similarly, Ahmadi-Javid (2012) presented the Entropic Value at Risk (EVaR), a risk measure intended to "optimize for the outcome at q% worst cases". EVaR was specifically designed to improve computational efficiency compared with other risk measures. Works in fairness have also intersected with robust solutions. Hashimoto et al. (2018) studied fairness without demographics, shifting the objective to minimize loss over any subset of samples, resulting in a robust optimization problem. Finally, Wang et al. (2024) considered a similar problem of min-max fairness without demographic information, leveraging variance as a proxy to minimize.

## 7   DISCUSSION

In this section, we discuss limitations of our work and future directions.

**Multi-class Classification**   Our results were exclusive for binary classification. While multi-class classification is a relevant task with wide applications, extending the results to this setting is non-trivial, as ambiguity would not result from a thresholding operation, but from the argmax operator. A similar idea of concentration could be employed in this setting by considering the event of the output probabilities being close to the uniform distribution.

**Sources of Multiplicity**   We restricted our analysis to multiplicity arising from stochasticity in the training procedure $\mathcal{T}$ (e.g., random seeds, splits). However, predictive multiplicity can arise from different steps of the standard machine-learning pipeline, including decisions during data pre-processing (Simson et al., 2024), architecture selection, and others. Despite the generality of our formulation, we do not consider other sources of predictive multiplicity, which is an interesting direction for future work.

**Interaction with Fairness**   While we motivated our work with potential trust issues arising from ambiguity, recent literature has also highlighted that unfairness mitigation procedures can increase ambiguity. Our theoretical results guide the reduction of higher-order error moments, resulting in a lower discrepancy between samples, which might positively affect fairness. Further investigation is needed to determine whether the proposed exponential loss, which focuses on hard samples, aligns with Min-Max/Rawlsian fairness objectives.

## 8   CONCLUSION

In this work, we studied the rate of disagreement among classifiers obtained from a randomized training procedure, known as ambiguity. We follow the approach of previous work, treating classifiers as approximations to a shared reference function, but we consider a broader class of classifiers. In this setting, ambiguity can be bounded by a term representing the concentration of mass around the threshold and another term related to the approximation quality. By focusing on this second term, we can bound ambiguity by the Brier score of models obtained from the training procedure and by an exponential function of classifier residuals. Based on our theoretical results, we propose a threshold selection methodology to reduce ambiguity that searches for thresholds at low concentrations, and an adapted loss for binary classification that emphasizes the penalty for large errors. On experiments with five datasets, both methodologies present a reduced ambiguity in comparison with standard training and ensembling. Furthermore, by modifying the training objective, we ensure that individual models are intrinsically robust to multiplicity, eliminating the dependency on post-hoc aggregation methods to achieve reliable decisions.

ACKNOWLEDGMENTS

This project was supported by the brazilian Ministry of Science, Technology and Innovations, with resources from Law nº 8,248, of October 23, 1991, within the scope of PPI-SOFTEX, coordinated by Softex and published Arquitetura Cognitiva (Phase 3), DOU 01245.003479/2024 -10. This study was financed, in part, by the São Paulo Research Foundation (FAPESP), Brazil. Process Number #2024/17292-9.

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

# A PROOFS

## A.1 PROOF OF PROP. 2.4

We can expand the definition of average ambiguity by applying Fubini's theorem to swap the expectations (valid as the integrand is non-negative):

$$
\mathbb{E}_{X\sim P_X}[A(X)] = \mathbb{E}_{X\sim P_X}\left[\mathbb{E}_{h,h'\sim U[\hat{\mathcal{R}}_m]}[1\{f_t(h(X)) \neq f_t(h'(X))\}]\right] \tag{14}
$$

$$
= \mathbb{E}_{h,h'\sim U[\hat{\mathcal{R}}_m]}\left[\mathbb{E}_{X\sim P_X}[1\{f_t(h(X)) \neq f_t(h'(X))\}]\right] \tag{15}
$$

$$
= \mathbb{E}_{h,h'\sim U[\hat{\mathcal{R}}_m]}[P_X(f_t(h(X)) \neq f_t(h'(X)))] \tag{16}
$$

$$
= \mathbb{E}_{h,h'\sim U[\hat{\mathcal{R}}_m]}[d(h,h')] \tag{17}
$$

## A.2 PROOF OF PROP. 3.1

The proof relies on decomposing the ambiguity based on the proximity of the predicted score $h(X)$ to the decision threshold $t$. If both $h(X), h'(X)$ are distant from the threshold (by a margin $\gamma$), a disagreement between models implies that at least one model has a large residual (error). Using the expansion of average ambiguity from Prop. 2.4:

$$
\mathbb{E}_{X\sim P_X}[A(X)] = \mathbb{E}_{h,h'\sim U[\hat{\mathcal{R}}_m]}[P_X(f_t(h(X)) \neq f_t(h'(X)))] \tag{18}
$$

Now, we analyze the inner probability term for a fixed pair of models $h, h'$. We decompose the event of disagreement based on whether at least one score $h(X), h'(X)$ is within a distance $\gamma$ of the threshold $t$. Let's write the events $A := (f_t(h(X)) \neq f_t(h'(X)))$, $B := (|h(x) - t| \leq \gamma \ \vee |h'(x) - t| \leq \gamma)$. We have:

$$
P_X(f_t(h(X)) \neq f_t(h'(X))) = P_X(A) = P_X(A \wedge \neg B) + P_X(A \wedge B) \tag{19}
$$

$$
\leq P_X(A \wedge \neg B) + P_X(B) \tag{20}
$$

$$
\leq P_X(A \wedge \neg B) + P_X(|h(x) - t| \leq \gamma) + P_X(|h'(x) - t| \leq \gamma) \tag{21}
$$

$$
= P_X(A \wedge \neg B) + 2S_\gamma(t) \tag{22}
$$

$$
\tag{23}
$$

We now focus on the first term, $P_X(A \wedge \neg B)$. If models disagree, each one should be on one side of the threshold. Furthermore, given that they are a distance of $\gamma$ from the threshold, there is at least a distance of $2\gamma$ between them.

$$
P_X(A \wedge \neg B) \leq P_X(h(X) \in [0, t-\gamma] \wedge h'(X) \in [t+\gamma, 1]) + P_X(h'(X) \in [0, t-\gamma] \wedge h(X) \in [t+\gamma, 1]) \tag{24}
$$

$$
\leq 2P_X(|h(X) - h'(X)| > 2\gamma) \tag{25}
$$

$$
= 2P_X(|h(X) - r(X) + r(X) - h'(X)| > 2\gamma) \tag{26}
$$

$$
\leq 2P_X(|h(X) - r(X)| > \gamma \vee |r(X) - h'(X)| > \gamma) \leq 2P_X(|h(X) - r(X)| > \gamma) \leq 2C_\gamma \tag{27}
$$

Recall the definition $C_\gamma := \max_{h \in \hat{\mathcal{R}}_m} P_X(|h(X) - r(X)| \geq \gamma)$. We now substitute this back into the expectation over $h, h'$. Since $h, h' \in \hat{\mathcal{R}}_m$, it holds that $P_X(|h(X) - r(X)| > \gamma) \leq C_\gamma$ for any $h$ drawn from the set.

$$
\mathbb{E}_{h,h'\sim U[\hat{\mathcal{R}}_m]}[P_X(f_t(h(X)) \neq f_t(h'(X)))]
$$

$$
\leq \mathbb{E}_{h,h'\sim U[\hat{\mathcal{R}}_m]}[2C_\gamma + 2S_\gamma(t)] = 2C_\gamma + 2S_\gamma(t) \tag{28}
$$

which concludes the proof.

### A.3 PROOF OF THEO. 3.2

We first present an auxiliary lemma.

**Lemma A.1.** *Consider a set of i.i.d. samples $(Z_1, \ldots, Z_n)$ from a random variable $Z$, with finite variance that are bounded by $[a, b]$. We have, with probability $1 - \delta$ :*

$$\frac{1}{n} \sum_{i=1}^{n} Z_i - E[Z] \le (b - a) \sqrt{\frac{\ln(1/\delta)}{2n}} \tag{29}$$

*Proof.* We start from the one-sided Hoeffding's inequality, stated for the sample mean $\bar{Z}_n$ (where $\bar{Z}_n = \frac{1}{n} \sum Z_i$ and $\mu = E[Z]$):

$$P(\bar{Z}_n - \mu \ge \epsilon) \le \exp\left(-\frac{2n\epsilon^2}{(b-a)^2}\right) \tag{30}$$

We want to find the value of $\epsilon$ such that this probability of error is bounded by $\delta$. We set the right-hand side to be our probability budget, $\delta$:

$$\delta = \exp\left(-\frac{2n\epsilon^2}{(b-a)^2}\right) \implies \tag{31}$$

$$\ln(\delta) = -\frac{2n\epsilon^2}{(b-a)^2} \implies \tag{32}$$

$$\ln\left(\frac{1}{\delta}\right) = \frac{2n\epsilon^2}{(b-a)^2} \implies \tag{33}$$

$$\epsilon^2 = \frac{(b-a)^2 \ln(1/\delta)}{2n} \implies \tag{34}$$

$$\epsilon = (b-a) \sqrt{\frac{\ln(1/\delta)}{2n}} \tag{35}$$

This $\epsilon$ is the specific value for which $P(\bar{Z}_n - \mu \ge \epsilon) \le \delta$. Therefore, the complementary event must hold with probability at least $1 - \delta$:

$$P\left(\bar{Z}_n - E[Z] \le (b-a) \sqrt{\frac{\ln(1/\delta)}{2n}}\right) \ge 1 - \delta \tag{36}$$

$\square$

We now present a proof of the main theorem. From the union bound we have that:

$$\mathop{P}_{h,h' \sim U[\hat{\mathcal{R}}_m]} \left( \bigcup_{x \in \mathcal{D}_{test}} f_t(h(x)) \ne f_t(h'(x)) \right) \le \sum_{x \in \mathcal{D}_{test}} \mathop{P}_{h,h' \sim U[\hat{\mathcal{R}}_m]} (f_t(h(x)) \ne f_t(h'(x))) \tag{37}$$

$$= \sum_{x \in \mathcal{D}_{test}} A(x) \tag{38}$$

Making $Z = A(X)$, where the randomness of $Z$ results from $P_X$ and using Lemma A.1, with $b = 1, a = 0$ (limits of $A(x)$), we have that:

$$\sum_{x \in \mathcal{D}_{test}} A(x) \le n \left( E_{x \sim P_X}[A(x)] + \sqrt{\frac{\ln(1/\delta)}{2n}} \right) \tag{39}$$

Finally, by Prop. 3.1 we can see that:

$$\mathop{P}_{h,h' \sim U[\hat{\mathcal{R}}_m]} \left( \bigcup_{x \in \mathcal{D}_{test}} f_t(h(x)) \ne f_t(h'(x)) \right) \le n \left( 2S_\gamma(t) + 2C_\gamma + \sqrt{\frac{\ln(1/\delta)}{2n}} \right) \tag{40}$$

## A.4   PROOF OF THEO. 3.3

Recall that $C_\gamma := \max_{h \in \hat{\mathcal{R}}_m} P_X(|h(X) - r(X)| \geq \gamma)$. For any fixed $h \in \hat{\mathcal{R}}_m$, we apply Markov's inequality to the squared residual $|h(X) - r(X)|^2$:

$$P_X(|h(X) - r(X)| \geq \gamma) = P_X(|h(X) - r(X)|^2 \geq \gamma^2) \leq \frac{\mathbb{E}_X[(h(X) - r(X))^2]}{\gamma^2} \tag{41}$$

Next, we consider the Brier score decomposition: $BS(h) = \mathbb{E}_X[(h(X) - r(X))^2] + \mathbb{E}_X[\text{Var}(Y|X)]$. Since the variance $\text{Var}(Y|X)$ is non-negative, it follows that $\mathbb{E}_X[(h(X) - r(X))^2] \leq BS(h)$. Substituting this into the inequality yields:

$$P_X(|h(X) - r(X)| \geq \gamma) \leq \frac{BS(h)}{\gamma^2} \tag{42}$$

Taking the maximum over $h \in \hat{\mathcal{R}}_m$ proves the theorem.

## A.5   PROOF OF THEO. 3.4

Recall the definition $C_\gamma := \max_{h \in \hat{\mathcal{R}}_m} P_X(|h(X) - r(X)| \geq \gamma)$. With a fixed $h$ and with $|h(X) - r(X)|$ as a random variable, we apply the Chernoff bound:

$$P_X(|h(X) - r(X)| \geq \gamma) = P_X(e^{s|h(X) - r(X)|} \geq e^{s\gamma}) \leq e^{-s\gamma} \mathbb{E}_X\left[e^{s|h(X) - r(X)|}\right] \tag{43}$$

The term $\mathbb{E}_X\left[e^{s|h(X) - r(X)|}\right]$ depends on the unobserved true probability $r(X)$. To relate this to the observed labels $Y$, we use Jensen's inequality. Consider the function $\phi(u) = e^{s|u|}$, which is convex. Note that $r(X) = \mathbb{E}_{Y|X}[Y]$. Thus:

$$|h(X) - r(X)| = |h(X) - \mathbb{E}_{Y|X}[Y]| = |\mathbb{E}_{Y|X}[h(X) - Y]| \tag{44}$$

By Jensen's inequality applied to the convex function $\phi(\cdot)$ and the conditional expectation $\mathbb{E}_{Y|X}$:

$$\phi(|\mathbb{E}_{Y|X}[h(X) - Y]|) \leq \mathbb{E}_{Y|X}[\phi(h(X) - Y)] \tag{45}$$

Substituting $\phi(u) = e^{s|u|}$:

$$e^{s|h(X) - r(X)|} \leq \mathbb{E}_{Y|X}\left[e^{s|h(X) - Y|}\right] \tag{46}$$

Taking the expectation over $X$ on both sides:

$$\mathbb{E}_X\left[e^{s|h(X) - r(X)|}\right] \leq \mathbb{E}_X\left[\mathbb{E}_{Y|X}\left[e^{s|h(X) - Y|}\right]\right] = \mathbb{E}_{X,Y}\left[e^{s|h(X) - Y|}\right] \tag{47}$$

Substituting this back into Eq. 43 completes the proof.

## A.6   PROOF OF THEO. 3.5

*Proof.* Following the result from Eq. 43 and the application of Jensen's inequality, we established that the tail probability is bounded by the exponential moment of the observed residuals:

$$C_\gamma \leq e^{-s\gamma} \mathbb{E}_{X,Y}\left[e^{s|h(X) - Y|}\right] \tag{48}$$

To provide a more interpretable bound that relies on standard error metrics (mean and variance of the residuals) and the finite support of the loss, we employ Bennett's inequality logic applied to the random variable $Z = |h(X) - Y|$. Note that since $h(X) \in [0, 1]$ and $Y \in \{0, 1\}$, $Z$ is bounded, i.e., $Z \in [0, M]$ (where typically $M = 1$).

Recall from Wainwright (2019) that the function $\phi(u) = \frac{e^u - 1 - u}{u^2}$ is non-decreasing for $u > 0$. Consequently, for any random variable $Z \in [0, M]$ and parameter $s > 0$, we have $sZ \leq sM$, which implies $\phi(sZ) \leq \phi(sM)$. Thus we have that:

$$\frac{e^{sZ} - 1 - sZ}{(sZ)^2} \leq \phi(sM) \tag{49}$$

$$e^{sZ} \leq 1 + \left(sZ + s^2 Z^2 \phi(sM)\right) \tag{50}$$

$$e^{sZ} \leq \exp\left(sZ + s^2 Z^2 \phi(sM)\right) \qquad \text{(since } 1 + x \leq e^x) \tag{51}$$

$$e^{sZ} \leq \exp\left(sZ + \frac{Z^2}{M^2}\left(e^{sM-1-sM}\right)\right) \tag{52}$$

Taking the expectation over the joint distribution of $X, Y$ on both sides of the inequality yields a bound on the MGF in terms of the first and second moments of the residual. Let $\mu = \mathbb{E}[Z]$ denote the expected absolute residual (MAE) and note that $\mathbb{E}[Z^2] = \mathbb{E}[(h(X) - Y)^2] = BS(h)$, which is the Brier score. Here, we will use the notation $m_2 := BS(h)$. Then:

$$E[\exp(sZ)] \leq \exp\left(sE[Z] + \frac{E[Z^2]}{M^2}(e^{sM} - 1 - sM)\right) = \exp\left(s\mu + \frac{m_2}{M^2}(e^{sM} - 1 - sM)\right) \tag{53}$$

Substituting this upper bound into the Chernoff inequality (Eq. 43), we minimize the exponent with respect to $s > 0$:

$$\ln(C_\gamma) \leq \min_{s>0}\left(-s\gamma + s\mu + \frac{m_2}{M^2}(e^{sM} - 1 - sM)\right) \tag{54}$$

Differentiating the objective function with respect to $s$ and setting it to zero to find the critical point:

$$-\gamma + \mu + \frac{m_2}{M^2}(Me^{sM} - M) = 0 \tag{55}$$

$$\frac{m_2}{M}(e^{sM} - 1) = \gamma - \mu \tag{56}$$

$$e^{sM} = 1 + \frac{M(\gamma - \mu)}{m_2} \implies s^* = \frac{1}{M}\ln\left(1 + \frac{M(\gamma - \mu)}{m_2}\right) \tag{57}$$

To simplify the final bound, let $u = \frac{M(\gamma-\mu)}{m_2}$, thus $s^* = \frac{1}{M}ln(1 + u)$, and define the function $g(u) = (1 + u)\ln(1 + u) - u$. Substituting $s^*$ back into the objective function yields:

$$\ln(C_\gamma) \leq -s^*\gamma + s^*\mu + \frac{m_2}{M^2}(e^{s^*M} - 1 - s^*M) \tag{58}$$

$$\leq -\frac{1}{M}ln(1 + u)(\gamma - \mu) + \frac{m_2}{M^2}(e^{\frac{1}{M}ln(1+u)M} - 1 - \frac{1}{M}ln(1 + u)M) \tag{59}$$

$$\leq -\frac{m_2}{M^2}ln(1 + u)u + \frac{m_2}{M^2}u - \frac{m_2}{M^2}ln(1 + u) = -\frac{m_2}{M^2}\left(ln(1 + u)(1 + u) - u\right) = -\frac{m_2}{M^2}g(u) \tag{60}$$

Thus, the probability of large residuals is bounded by:

$$C_\gamma \leq \exp\left(-\frac{E[Z^2]}{M^2}g\left(\frac{M(\gamma - E[Z])}{E[Z^2]}\right)\right) \tag{61}$$

This concludes the proof, demonstrating that $C_\gamma$ is controlled by the scale $M$, the mean absolute error, and the raw second moment of the residuals. $\qquad\square$

*Remark* A.2 (Impact of Moment Minimization on Ambiguity). The derived bound explicitly characterizes how the distribution of residuals controls the probability of large deviations $C_\gamma$. Analyzing the exponent $\mathcal{E}(m_2) = -\frac{m_2}{M^2}h\left(\frac{M(\gamma-\mu)}{m_2}\right)$ reveals the mechanism for reducing ambiguity:

- **Decreasing the Second Moment** ($m_2 = \mathbb{E}[Z^2]$)**:** As the Brier score decreases ($m_2 \to 0$), the argument of the Bennett function becomes large. In this regime, the bound behaves asymptotically as $C_\gamma \leq \exp\left(-\frac{\gamma-\mu}{M}\ln\left(\frac{1}{m_2}\right)\right)$. Since the ambiguity bound is exponential in this term ($C_\gamma \leq \exp(-\mathcal{E})$), this logarithmic growth in the exponent translates to a fast polynomial decay of ambiguity with respect to the Brier score ($C_\gamma \propto m_2^k$). This confirms that minimizing the second moment is a highly efficient strategy for suppressing disagreement.

- **Decreasing the Mean Absolute Error** ($\mu = \mathbb{E}[Z]$)**:** A lower $\mu$ (assuming $\mu < \gamma$) increases the argument $u = \frac{M(\gamma - \mu)}{m_2}$, thereby decreasing the bound. However, this effect is bounded: as $\mu \to 0$, the numerator converges to a constant $M\gamma$, limiting the potential gain compared to the unbounded improvements possible via variance reduction.

- **Decreasing the Maximum Scale** ($M$)**:** Decreasing the support $M$ would theoretically produce the largest effect on the bound by restricting worst-case deviations. However, strictly controlling the absolute tail of the distribution is typically hard to attain. While explicitly minimizing higher-order moments is possible, it often introduces numerical instability; therefore, the practical lever for optimization is to prioritize the second moment ($m_2$), maintaining only a minor emphasis on higher moments to ensure robustness.

## B  DATASETS

We leverage widely employed benchmark datasets of classification from different contexts, such as marketing, finance, and social domains. Most of the used datasets were obtained from the UCI Machine Learning repository (Kelly et al.). At Tab. 2, we present a summary of datasets' descriptions. In more detail, the datasets were:

- **Taiwan Credit**: This dataset contains credit card default data of clients from a Taiwanese bank. The objective is to predict if a client will default on the next month based on features such as age, gender, and payment data from previous months.

- **Adult**: Adult dataset is obtained from a US Census of 1994, collecting socioeconomic features of individuals, such as age, gender, marital-status, occupation, education-level, and if the individual has a yearly income higher than $50K.

- **ACSIncome**: This dataset was presented by Ding et al. (2021) with the objective of replacing the Adult dataset in fairness studies. It is a larger collection of census data from the US. Features included are gender, race, occupation, with the label being if the individual has a yearly income higher than $50K. We leverage data from all states for the year 2018, being the biggest dataset considered.

- **BankMarketing**: This dataset was collected by a Portuguese banking institution during a marketing campaign based on phone calls. The objective is to predict if the client will make a subscription deposit based on the features such as age, marital status, job, and variables about previous contact of the bank with the individual.

- **ENEM**: This dataset is obtained from a national high-school exam applied in Brazil. The task is to predict the score on the test based on socio-economic features, such as the type of school where studied (public, private), occupancy of family members, and others. 50,000 samples were selected from the year of 2018. The preprocessing was the same as the work by Alghamdi et al. (2022).

Table 2: Summary of Dataset Statistics.

| Dataset | Size ($N$) | Features | Categorical | Preprocessed features | $P(Y = 1)$ |
|---|---|---|---|---|---|
| BankMarketing | 45,211 | 13 | 8 | 29 | 0.117 |
| Adult | 48,842 | 12 | 8 | 71 | 0.161 |
| Taiwan | 30,000 | 23 | 3 | 30 | 0.221 |
| ACSIncome | 50,000 | 11 | 8 | 79 | 0.370 |
| ENEM | 50,000 | 148 | 148 | 161 | 0.499 |

## C  EXTRA EXPERIMENT DETAILS

Default hyperparameters used by procedure $\mathcal{T}_{split}$ and hyperparameter spaces used by procedure $\mathcal{T}_{tune}$ are presented in Tab. 3.

Table 3: Model Hyperparameters and Search Spaces.

| Model | Hyperparameter | Default Value | Search Space |
|-------|----------------|---------------|--------------|
| MLP | Hidden Layer Sizes | 64 | $\{32, 64, 128\}$ |
| | N. layers | 2 | $\{1, 2, 3\}$ |
| | Learning Rate | 0.001 | $[0.001, 0.1]$ (log-uniform) |
| | Weight Decay | $10^{-4}$ | $[10^{-6}, 10^{-2}]$ (log-uniform) |
| LGBM | Number of Estimators | 100 | $[5, 250]$ |
| | Learning Rate | 0.1 | $[0.05, 1.0]$ |
| | Max Depth | None | $[2, 12]$ |

## D  CONNECTION TO GENERALIZED EVAR AND MOMENT BOUNDS

From Theorem 3.4, we observe that for a given deviation $\gamma$, controlling the tail of the residual distribution is equivalent to finding the optimal tilting parameter $s^*$ in the Chernoff bound:

$$P(|h(x) - r(X)| \geq \gamma) \leq \inf_{s>0} e^{-s\gamma} \mathbb{E}\left[e^{s|h(x)-r(X)|}\right].$$

This involves minimizing the Moment Generating Function (MGF) of the residuals. While the critical point condition yields an implicit relationship between $\gamma$ and $s^*$, specifically $\gamma = \frac{\mathbb{E}[Ze^{s^*Z}]}{\mathbb{E}[e^{s^*Z}]}$ where $Z = |h(x) - r(X)|$, relying solely on this requires prior knowledge of the full residual profile.

To provide a more principled understanding of how minimizing the tilted objective relates to controlling specific properties of the distribution (such as higher moments and worst-case scenarios), we analyze its connection to the Entropic Value at Risk (EVaR) and Bennett's Inequality.

**EVaR.**  The Entropic Value at Risk (EVaR) provides the tightest upper bound on the tail probability via the Chernoff inequality. For a confidence level $\alpha$, the dual representation of EVaR finds the minimal threshold $\gamma$ such that the probability bound holds:

$$\text{EVaR}_{1-\alpha}(Z) := \inf_{s>0} \frac{1}{s} \ln\left(\frac{\mathbb{E}\left[e^{sZ}\right]}{\alpha}\right).$$

Minimizing the tilted loss is thus intimately connected to minimizing the EVaR of the residuals. We now show that this framework can be generalized to control higher-order moments of the residuals.

**Theorem D.1** (Generalized EVaR for Higher Moments). *Let $X \geq 0$ be a random variable. The concept of EVaR can be generalized to bound the tail of the $k$-th moment $X^k$ for $k > 0$:*

$$EVaR_{1-\alpha}(X^k) \equiv \inf_{r>0}\left[\frac{1}{r}\ln\left(\frac{\mathbb{E}[e^{rX^k}]}{\alpha}\right)\right].$$

*Alternatively, formulated in terms of the original variable scale, we define the generalized risk measure:*

$$EVaR_{1-\alpha}(X) \equiv \left[\inf_{r>0}\frac{1}{r}\ln\left(\frac{\mathbb{E}[e^{rX^k}]}{\alpha}\right)\right]^{\frac{1}{k}}, \quad \forall k > 0.$$

*Proof.* Consider the transformation functions $f(u) = u^k$ and $g(v) = e^v$. Since both $f$ and $g$ are non-decreasing functions for $u \geq 0$ and $k > 0$, applying the monotonicity property of probability and Markov's Inequality yields:

$$P(X \geq \gamma) = P(X^k \geq \gamma^k) = P(e^{sX^k} \geq e^{s\gamma^k}) \leq e^{-s\gamma^k}\mathbb{E}[e^{sX^k}].$$

Setting the right-hand side equal to the confidence level $\alpha$ and solving for $\gamma$, we obtain the bound:

$$\gamma^k \leq \frac{1}{s}\ln\left(\frac{\mathbb{E}[e^{sX^k}]}{\alpha}\right).$$

Minimizing this upper bound over $s > 0$ recovers the definition of EVaR applied to the random variable $X^k$. Since the event $\{X \geq \gamma\}$ is equivalent to $\{X^k \geq \gamma^k\}$, finding the minimal $\gamma$ is equivalent to minimizing the EVaR of the moment $X^k$ and taking the $k$-th root:

$$\text{EVaR}_{1-\alpha}(X) \leq \left( \inf_{s>0} \frac{1}{s} \ln \left( \frac{\mathbb{E}[e^{sX^k}]}{\alpha} \right) \right)^{1/k}.$$

$\square$

*Remark* D.2 (Stability and Moment Control). The generalization to higher moments is not merely theoretical; it offers significant numerical advantages. The standard exponential moment ($k = 1$) can be highly sensitive to the tail behavior of residuals, leading to unstable optimization landscapes. By formulating the bound in terms of $X^k$, specifically for $k = 2$, we can leverage more stable estimates of second-order moments (variance) while when $s \to 0$ it converges to $\frac{BS}{\alpha}$. This connection allows us to bridge the gap between Entropic Risk and concentration inequalities that rely on variance, such as Bennett's inequality discussed previously.

Worth mentioning that, despite being based on EVaR, our approach fixes $s$ optimizes only $\frac{1}{s} ln \left( \mathbb{E}[e^{sX^k}] \right)$ since other variables remain constant.

