# OpenReview forum: "Navigating the Rashomon Set: The Impact of Score Distributions and Decision Thresholds on Model Agreement"
_ICLR.cc/2026/Workshop/AFAA — AFAA 2026 Poster_

### Official Review · Reviewer_yd1E · 2026-02-19
**Interesting problem, good use of high-dim prob theory, weak use and discussion of algorithm**

**Rating:** 2
**Confidence:** 5

**Summary:**

In this article, the authors consider the problem of multiple models, which has been referred to as the Rashomon set in the recent literature.  This paper suggests developing metrics for exploring and evaluating the models.  The major premise of this paper is to focus on how different models might make predictions for a given data point.  Based on this, the authors define an average ambiguity and define an upper bound that is a combination of a so-called concentration measure as well as the tail behavior of a `random residual’ (Equation (5) in the paper).   A series of bounds for the concentration measure is presented that reflect tools used commonly in high-dimensional probability theory.  The approach the authors take is to consider multiple thresholds and to perform an optimization (equation (10) in the paper).  A series of experiments are performed to evaluate the ambiguity of a given dataset.

**Strengths:**

The authors make use of some very sensible techniques from high-dimensional probability theory and do a nice job of summarizing some relevant literature.    Also, this problem of how to better understand different models that predict well but which might have ambiguity in predictions for individuals.

**Weaknesses:**

I have a whole slew of questions that limit my enthusiasm about the work:

1.	In equation (10), there is no definition provided for S(t) (without a subscript). I also do not have a sense of the optimization problem complexity represented in (10).  It is not obvious to me how the optimization algorithm works, which is at the heart of the entire paper.
2.	How informative are bounds such as (7) – (9)?   Relatedly, what are values of gamma analysts would be choosing in practice?
3.	I think that Corollary 3.2. is fairly obvious and should not rise to a corollary result.
4.	The measure S_{\gamma}(t) is intimately tied to the margin/low-source noise condition in the machine learning theory literature but was not addressed at all by the authors.
5.	Given that the work appears to be based on behavior for a given sample datapoint, I do wonder if the explainable ML literature wouldn’t be relevant here as well.
6.	 I would liked to have seen a greater discussion of the implications of the results in Table 5.  This seems to suggest that the choice of loss function matters, which the authors take as a given.  Conversely, it also might speak to the inherent `easiness/difficulty’ of certain classification problems, where the bank marketing might be easier and the ACS income one harder.

---

### Official Review · Reviewer_WVxW · 2026-02-21
**Good workshop paper but not immediately on the workshop topic**

**Rating:** 4
**Confidence:** 4

**Summary:**

The paper theoretically connects model ambiguity within the Rashomon set to the distribution of prediction residuals and decision thresholds. Based on this grounding, the authors propose a simple threshold selection method and a tilted loss function that effectively reduce pairwise model disagreement.

**Strengths:**

There is a clear improvement over ensembling, which is a common baseline method for handling multiplicity.

The paper has an intuitive but reasonable theoretical grounding that directly connects ambiguity to model errors.

The proposed procedures like the threshold selection and tilted loss are accessible and reasonable to implement in standard pipelines.

The paper provides good empirical results that validate the theoretical work.

**Weaknesses:**

There is no clear connection to fairness, which is the central theme of the workshop. By changing the threshold while trying to resolve ambiguity, fairness can actually become worse because it might disproportionately affect certain subgroups. A more formal trade-off discussion would make the paper significantly stronger.

Please discuss how the definitions of ambiguity and dataset ambiguity are related to the pairwise disagreement metric defined by Black et al.

It would be interesting to quantify the cost in the ambiguity-accuracy trade-off based on dataset properties (noise, class imbalance, etc.), which seems to vary based on Figure 3. While the results stay in the Rashomon set, exploring this connection would make for a really interesting discussion to add to the paper.

This is an interesting workshop paper, so I will recommend acceptance if the AC is willing to accept a paper that is not fairness-first for the workshop.

---

### Official Review · Reviewer_nnP4 · 2026-02-23
**Reducing Rashomon ambiguity via residual-tail control and practical threshold selection**

**Rating:** 4
**Confidence:** 4

**Summary:**

This paper studies predictive multiplicity (the “Rashomon set”) in binary classification through the lens of model agreement under decision thresholds. The authors introduce an “ambiguity” measure capturing the extent to which different high-performing models disagree on individual predictions. They provide theoretical results linking ambiguity to the distribution of residuals especially the frequency of large residuals and propose (i) a simple data-driven approach for choosing thresholds that reduces ambiguity with minimal accuracy loss and (ii) a modified (“tilted”) objective / loss that explicitly down-weights large-residual behavior to shrink disagreement. Experiments on five real datasets demonstrate reductions in ambiguity at modest cost in accuracy and compute.

**Strengths:**

- Clear problem framing: Addresses a real pain point in deployment, multiple “good” models leading to inconsistent decisions, rather than treating multiplicity as purely a philosophical issue.
- Conceptual contribution: The link between ambiguity and the tail behavior of residuals gives an interpretable diagnostic for when/why disagreement is large.
- Actionable methods: Provides two concrete mitigations (threshold selection and loss/objective adjustment) that can be implemented without changing the overall model class.
- Empirical breadth: Uses five datasets and compares across common model families, helping demonstrate that the effect is not a single-dataset artifact.

**Weaknesses:**

- Ambiguity definition sensitivity: Ambiguity can depend strongly on (i) how the Rashomon set is operationalized (training randomness vs explicit constraint set) and (ii) the chosen accuracy band. It would help to include a sensitivity analysis over the Rashomon “tolerance” and show how stable the conclusions are.
- Thresholding vs calibration: Since thresholds interact with calibration, it would strengthen the story to report calibration metrics (e.g., ECE) and/or show whether ambiguity reductions persist after calibration (Platt/isotonic) or under cost-sensitive settings.
- Generalization beyond tabular binary tasks: The approach is demonstrated on tabular datasets; a short discussion (or small experiment) on how this extends to multiclass or distribution shift would make the contribution feel more broadly applicable.
- Practical guidance: The threshold selection method would benefit from a clearer “recipe” (hyperparameters, validation protocol, computational overhead) so a practitioner can reproduce the ambiguity–accuracy trade-off reliably.
- Ablations: A tighter ablation separating (a) threshold choice alone, (b) tilted objective alone, and (c) both combined under matched training budgets, would make the empirical takeaway crisper.

---

### Meta-Review · Area_Chair_rCDH · 2026-02-24

**Recommendation:** Main Papers Track
**Confidence:** 4

**Metareview:**

Three reviewers have evaluated the paper with mixed but overall positively leaning conclusions (summary assessment: accept/accept/reject).

The review value the practical, impact-oriented motivation of the work with a focus on an important problem in ML-based decision-making, along with its proposed actionable solutions. The paper presents a novel conceptual contribution to multiplicity research and a strong experimental evaluation. However, please note that some of the chosen datasets such as Adult are contested in the fair ML community and a more principled selection of data could have improved the generalizability of the experiments.

At the same time, it is noted that the paper primarily focuses on binary prediction with tabular data, which one could see as a bit at odds with the workshop theme, particularly as the link to algorithmic fairness is not well developed as well.
The reviews propose to add further evaluation metrics such as calibration, clarification of the optimization algorithm, as well as added pointers to further related literature, which I believe would be feasible minor revisions. I would further argue that the multiplicity problem is genuinely linked to fairness, which establishes the relevance for the workshop.

I therefore propose the paper to be accepted at this stage.

---

### Decision · Program_Chairs · 2026-03-02

Accept (Poster)